# Phylogenetic and Structure-Function Analyses of ENA ATPases: A Case Study of the ENA1 Protein from the Fungus *Neurospora crassa*

**DOI:** 10.3390/ijms25010514

**Published:** 2023-12-30

**Authors:** Marcos Aguilella, Blanca Garciadeblás, Luis Fernández Pacios, Begoña Benito

**Affiliations:** 1Centro de Biotecnología y Genómica de Plantas (CBGP), Universidad Politécnica de Madrid (UPM), Instituto Nacional de Investigación y Tecnología Agraria y Alimentaria (INIA), Campus Montegancedo UPM, Pozuelo de Alarcón, 28223 Madrid, Spain; marcos.aguilella.3n@gmail.com; 2Departamento de Biotecnología-Biología Vegetal, Escuela Técnica Superior de Ingeniería Agronómica, Alimentaria y de Biosistemas, Universidad Politécnica de Madrid (UPM), 28040 Madrid, Spain; b.garciadeblas@upm.es

**Keywords:** ENA ATPases, Na^+^ transporters, Alphafold2, *Neurospora crassa*, phylogeny

## Abstract

ENA transporters are a group of P-type ATPases that are characterized by actively moving Na^+^ or K^+^ out of the cell against their concentration gradient. The existence of these transporters was initially attributed to some fungi, although more recently they have also been identified in mosses, liverworts, and some protozoa. Given the current increase in the number of organisms whose genomes are completely sequenced, we set out to expand our knowledge about the existence of ENA in organisms belonging to other phylogenetic groups. For that, a hidden Markov model profile was constructed to identify homologous sequences to ENA proteins in protein databases. This analysis allowed us to identify the existence of ENA-type ATPases in the most primitive groups of fungi, as well as in other eukaryotic organisms not described so far. In addition, this study has allowed the identification of a possible new group of P-ATPases, initially proposed as ENA but which maintain phylogenetic distances with these proteins. Finally, this work has also addressed this study of the structure of ENA proteins, which remained unknown due to the lack of crystallographic data. For this purpose, a 3D structure prediction of the NcENA1 protein of the fungus *Neurospora crassa* was performed using AlphaFold2 software v2.3.1. From this structure, the electrostatic potential of the protein was analyzed. With all these data, the protein regions and the amino acids involved in the transport of Na^+^ or K^+^ ions across the membrane were proposed for the first time. Targeted mutagenesis of some of these residues has confirmed their relevant participation in the transport function of ENA proteins.

## 1. Introduction

The ability to exchange molecules with the environment and respond to changes in environmental conditions is vital for the survival of any organism. This property is performed by a series of receptor and/or ion- or molecule-transporting proteins that are immersed in lipid bilayers. One of the protein families that provides the capacity to transport ions across cell membranes is the so-called P-type ATPases, a family of transporters that uses the energy of ATP hydrolysis to transport cations such as Na^+^, K^+^, Ca^2+^, Cu^2+^, or H^+^, among others, and phospholipids against their concentration gradient [1]. The designation of P-type ATPases derives from the fact that during the catalytic cycle of protein activity, a transient intermediate is formed by phosphorylating an aspartate residue [2]. Traditionally, the transport cycle is described very simplistically according to the classical Albers-Post model [3,4], in which these proteins are considered to undergo major conformational changes coupled to ion translocation. According to this model, P-ATPases can be found in two distinct conformational states: E1 and E2, each showing different affinity for the nucleotide and the transported ion.

P-type ATPases are relatively large, ubiquitous, and diverse membrane proteins involved in many transport processes in virtually all living organisms. They are characterized by sharing a highly conserved structural organization and a set of sequence motifs involved in their mechanism of action and function. The Ca^2+^-ATPase SERCA1a (from Sarco (Endo)plasmic Reticulum CAlcium) from rabbit muscle was the first P-type ATPase in which its crystal structure was determined [5], and today we have extensive information on the structure of this pump in various conformations [6]. Later, structures of other P-type pumps also emerged, including the Na^+^/K^+^-ATPase [7] and the plasma membrane H^+^-ATPase [8]. Despite a relatively low degree of sequence conservation, the structures of these pumps are strikingly similar, which allows the reliable prediction of model structures of P-ATPases on the basis of available experimental structural information. P-type ATPases are multidomain proteins that feature a transmembrane domain consisting of between 6 and 13 transmembrane (TM) helices and three cytoplasmic domains: the A domain (Actuator), the N domain (Nucleotide binding), and the P domain (Phosphorylation domain) [6,9]. The A domain is located between the TM2 and TM3 helices and forms a flexible globular domain that allows it some rotation. This domain contains one of the highly conserved sequence motifs of P-ATPases, the TGES motif, which stabilizes the transition between the two conformational states and participates in the hydrolytic dephosphorylation of aspartate mediated by a water molecule [9]. The P domain is located between the TM4 and TM5 helices and contains the conserved motif DKTG, whose first amino acid is the aspartate residue that is phosphorylated. Other conserved motifs in this domain are DPPR, TGDN, and GDGxND, which are involved in Mg^2+^ coordination, ATP binding, and aspartate phosphorylation. Finally, the N domain is inserted in the middle of the P domain and is responsible for ATP binding and phosphorylation of the P domain. This domain contains the conserved KGAPE motif [1].

Based on sequence homology, phylogenetic studies of P-type ATPases have identified five subfamilies (named P1-P5), characterized by substrate specificity and sequence motifs specific to each group: The P1 subfamily transports metals, such as Cu^2+^ or Cd^2+^; the P2 subfamily includes Ca^2+^, Na^+^, Mg^2+^ or K^+^ transporters; the P3 subfamily includes H^+^ ATPases; the P4 subfamily transports phospholipids, being involved in the maintenance of lipid membrane asymmetry; and, finally, the P5 subfamily, whose proteins seem to be involved in the regulation of cation homeostasis in the endoplasmic reticulum [10,11], although their specificity is still unknown. Subfamilies P1, P2, P3, and P5 are further subdivided into eleven additional classes [10,12].

Within the subfamily of P2 ATPases are the so-called ENA-type proteins. These transporters belong to the P2D class, closely related to the endoplasmic reticulum (P2A), membrane (P2B) Ca^2+^ pumps, and the Na^+^/K^+^ pumps (P2C). ENA (from *Exitus NAtru*, Latin for “sodium exit”) ATPases were first discovered in the yeast *Saccharomyces cerevisiae* [13], where their activity and substrate specificity were determined [14]. Unlike the Na^+^/K^+^ of the P2C group, which exchanges Na^+^ (outward) for K^+^ (inward), ENA ATPases are characterized by their capacity to transport Na^+^ and/or K^+^ to the cell exterior, thus relieving an excessive accumulation of these cations in the cytoplasm. Until now, the lack of experimental structural data on ENA proteins have prevented us from gaining a deeper understanding of the molecular mechanisms of their functions.

ENA proteins can actively extrude Na^+^ (and/or K^+^) against their concentration gradient, which confers to their host organisms the ability to survive in saline environments, even in alkaline pH conditions, where other Na^+^ transport mechanisms such as Na^+^-H^+^ antiporters fail, being dependent on an H^+^ concentration gradient. The existence of these transporters was initially attributed to some fungi, although more recently they have also been identified in mosses, liverworts, and some protozoa. However, ENA proteins are not found in higher organisms such as plants or animals, which could potentially make them a good target for inhibitory drugs to control pathogenic protozoa and fungi. Systematic functional studies of different fungal ENA ATPases have shown that some ENA ATPases are specific for Na^+^ efflux (such as NcENA1 from *Neurospora crassa* [15]), while others show poor discrimination between Na^+^ and K+ or even that some might be specific for K^+^ efflux (the best known example of the latter is CTA3 from *Schyzosaccharomyces pombe* [16,17]. Transcriptional studies confirm that these proteins are expressed in the presence of high salt concentrations and at an alkaline pH [17]. By obtaining yeast defective mutants in the genes encoding these proteins, it has been possible to demonstrate that ENA proteins are essential for organisms that live exposed to saline environments, maintaining a low cytoplasmic concentration of Na^+^ to avoid its toxic effect and thus allowing their survival [18]. Saline media are both salinized soils widespread on the planet and marine environments, where a NaCl concentration of 500 mM is reached, as well as the plasma serum in which animal cells bathe (150 mM NaCl). This means that the activity of ENA proteins allows the proliferation and survival of organisms that possess them in extensive and very diverse environments.

ENA-type ATPases are widely distributed among fungi, although homologous proteins have also been identified in a group of primitive plants such as mosses (e.g., *Physcomitrella patens* [19]) and liverworts (*Marchantia polymorpha*), as well as in various protozoa such as *Trypanosoma* or *Leishmania*. However, ENA genes have not been found in higher organisms, such as vascular plants or animal cells [17]. This fact allows exploring the possibility of a dual biotechnological application of ENA proteins: as a target of action of drugs inhibiting their activity for the control of pathogenic protozoa and fungi [20], as well as using this type of transporter to increase tolerance to salinity in plant cultures through their heterologous expression [17]. On the other hand, the identification of ENA proteins in such diverse groups of organisms makes it interesting to conduct an in-depth phylogenetic study to find out the extent of these proteins in evolution. Pursuing these goals justifies extending the field of research on this group of transporters.

The aim of this work has been to perform a phylogenetic study of ENA proteins to update their evolutionary coverage. This study has been carried out by building a profile of a hidden Markov model and Bayesian inference. In addition, the prediction of the three-dimensional structure of the NcENA1 protein of the fungus *N. crassa* using AlphaFold2 has been addressed. Finally, a more exhaustive study of the structure and function of this protein has been conducted, trying to propose possible relevant amino acids involved in the formation of the specific channel through which Na^+^ and/or K^+^ are transported. By site-directed mutagenesis of some of the proposed amino acids and expression of the mutated ENA protein in yeast, we have been able to verify their role in protein function.

## 2. Results

### 2.1. Identification of ENA Proteins in New Groups of Organisms Using the Hidden Markov Model

To precisely determine the groups of organisms that possess ENA-type P-ATPases, an exhaustive search of these proteins in the currently available databases were proposed. For this purpose, a profile of a hidden Markov model defining the conserved regions of ENA proteins was constructed (see Materials and Methods). HMM profiles are a description of the consensus sequence of an MSA. These profiles use a position-specific scoring system to capture information about the degree of conservation at various positions in the multiple alignments. This fact makes it a much more sensitive and specific method for searching homologous sequences in databases than pairwise methods, such as those used by BLAST, which use position-independent scoring [21]. Once the HMM profile for ENA proteins was proven, it was used to search for novel ENA sequences by pitting the HMM profile against protein sequence databases such as the Joint Genome Institute (JGI) (https://genome.jgi.doe.gov/portal/pages/tree-of-life.jsf, v8.18.170, last accessed 18 October 2023), NCBI (https://www.ncbi.nlm.nih.gov/, last accessed 18 October 2023) [22], and Uniprot (https://www.uniprot.org/, last accessed 18 October 2023) [23].

The analysis corroborated what was already known: that ENA proteins do not exist in prokaryotic organisms and that the existence of these proteins is limited to some phylogenetic groups of eukaryotes. Among eukaryotes, new ENA proteins were identified within the fungal group (Figure 1 and Appendix A).

Although the existence of ENA proteins in this group was already known, the present study has allowed the identification of new homologous proteins that exist in phylogenetically more primitive groups of fungi, such as some belonging to the phylum Chytridiomycota and Zoopagomycota. Nonetheless, there are organisms currently classified within the group of fungi, such as those included in Cryptomycota and Microsporidia, as well as a group of the phylum Chytridiomycota classified as Chytridiomycetes, that do not seem to possess ENA ATPases. On the other hand, the search for ENA proteins within the Viridiplantae group confirmed the existence of ENA proteins in the Chlorophyta group and in primitive plants such as mosses and liverworts other than those previously identified, although they do not exist in Anthoceros, a group phylogenetically very close to both. This search has also confirmed that ENA proteins do not exist in higher plants. The non-identification of ENA ATPases in some organisms could be due to an incomplete or inappropriate annotation of the proteomes of these organisms, but it could rather indicate that throughout evolution and probably depending on the environmental conditions in which each organism has thrived, it has been selected to either have or not have certain mechanisms that fulfill specific physiological functions. Interestingly, the screening of ENA ATPases in other phylogenetic groups of eukaryotes showed their existence in new organisms belonging to the groups Heterobolosea, Heterokonta (also known as stramenopiles), and Rhizaria (Figure 1 and Appendix A).

From the results obtained in the search for proteins homologous to ENA ATPases, it was noteworthy that, although ENA representatives appeared within the phylum Euglenozoa, which includes the protozoa of genera *Trypanosoma*, *Leishmania*, and *Naegleria* that were already known to possess these proteins (Figure 1 and [17]), no homologues appeared in the pathogenic protozoan *Plasmodium*. This result was striking because it has recently been published that the *Plasmodium falciparum* parasite possesses a Na^+^ transporting ENA-type ATPase called PfATP4, which, in fact, is currently being intensively investigated as a possible target of action against this pathogen [24]. To clarify this discrepancy, *Plasmodium* proteins were subjected to ENA protein HMM profiling along with other proteins identified in other organisms that showed homology to that of *Plasmodium*, which had also recently been proposed as ENA-type ATPases [25] (included in Appendix A). Surprisingly, the HMM profile returned a non-zero ‘e-value’, greater than 8.7 × 10^−212^, which seemed to suggest that none of these proteins were ENA ATPases.

**Figure 1 ijms-25-00514-f001:**
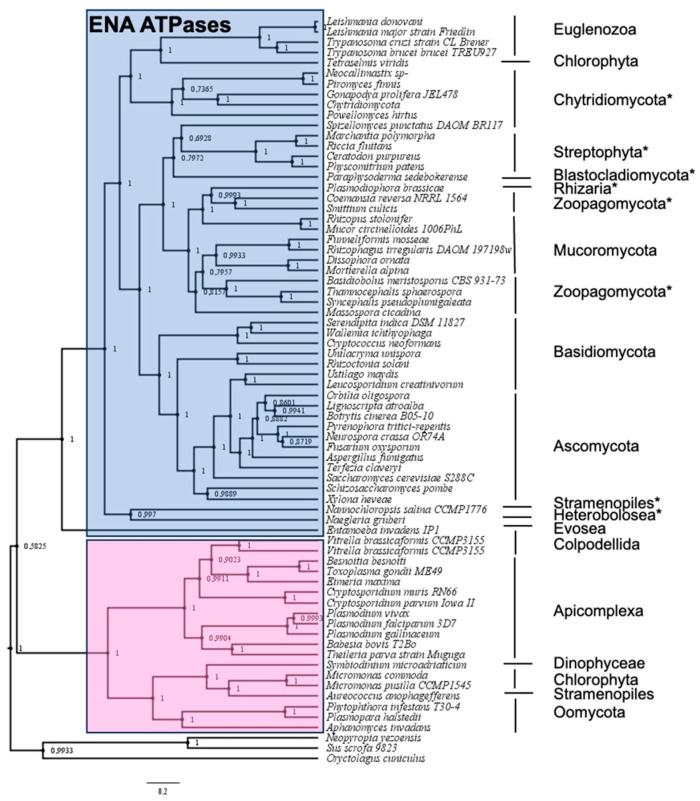
Phylogenetic tree of ENA ATPases. It was based on the alignment of 72 sequences compiled in Appendix A (Appendix A), which include known and newly identified ENA ATPases using the Hidden Markov Model (labeled with blue background), along with ‘proposed as’ ENA proteins in [25] (labeled with pink background). Three P2-type P-ATPases other than ENA proteins are included as an outgroup. Phylogenetic groups in which ENA ATPases have been identified for the first time are marked with asterisks. Numbers at nodes indicate the probability of Bayesian inference. The phylum to which the different organisms belong is indicated on the right.

### 2.2. Phylogenetic Analysis of ‘Newly Identified’ and ‘Proposed as’ ENA ATPases

To find out whether the proposed P-ATPases of *Plasmodium* as well as those published in [25] were truly ENA ATPases, a phylogenetic comparison with other subfamilies of P-type ATPases (P1 to P5) was performed (Figure 2 and Appendix A, which contain information on the proteins chosen for comparison).

Special attention was paid to the choice of proteins representing the P2 subfamily in addition to the ENA ATPases (2D), such as the Ca^2+^-SERCA ATPases (2A) and the Na^+^/K^+^ ATPases (2C), with which the questioned ATPases could have more sequence similarity. In the selection of ATPases from the different subfamilies to compare, it was also considered to choose representatives belonging to protist organisms to reduce the bias derived from the organism from which the proteins originate. The inferred phylogenetic tree showed the existence of a new clade including the proteins initially proposed as ENA ATPases that is perfectly differentiated from the ENA and from the rest of the P-type ATPases clades (Figure 2). Interestingly, the location within the clade of ENA proteins of proteins from protist organisms that belong to the Euglenozoa, Stramenopiles, Heterobolosea, and Evosea groups allows us to rule out that the separation into two clades is due to the evolutionary separations that exist between these eukaryotic organisms and fungi.

If we compare the phylogenetic distances between the proteins involved in Na+ transport, i.e., ENA ATPases, Na^+^/K^+^ ATPases, and those forming the new clade [24,25], we observe that the latter are closer to ENA-type ATPases than to Na^+^/K^+^ ATPases. Therefore, it should be noted that, within the group of protists, there are organisms that possess typical ENA ATPases that are located in the ENA ATPase clade (sequences in Figure 2 numbered from 44 to 46) and others that possess a new type of P-ATPases that would form a separate clade (Figure 2, from 22 to 40), which would allow extending the diversity of P-ATPases with Na^+^ transport capacity.

### 2.3. Prediction of NcENA1 ATPase Structure

To have a model of the 3D structure of ENA ATPases, the prediction was made with AlphaFold2 software v2.3.1, using as an example the sequence of the ENA1 protein from the fungus *Neurospora crassa* (NcENA1). The complete sequence of this protein has 1121 amino acids and is known to be an efficient Na+ efflux system, although it also transports K^+^ to the outside with lower efficiency [15,17].

Among the five structure models provided by Alphafold2 for the NcENA1 protein, the one with the highest overall pLDDT value, 83.3%, was chosen, revealing high confidence in the model [26,27] (Appendix A). The comparison of this model with that included in the AFPSDb is shown in Appendix A. According to that recommended by the AlphaFold authors for end protein segments displaying especially low pLDDT values [26], the two short disordered segments N-terminal 1-6 and C-terminal 1084–1121 (that have per-residue pLDDT < 30%) were discarded in both models. Not only does the structural alignment demonstrate their close agreement (RMSD = 1.1 Å over the 1077 residues, Appendix A), but also their per-residue pLDDTs along the sequence reveal that they refer to the same structure (Appendix A). It is worth mentioning that our model obtained with AlphaFold2 shows in many residues pLDDT values even slightly better than those of the corresponding residues in the AFPSDd model (Appendix A). In fact, the number of residues with pLDDT values > 90% (confidence “very high”) is 466 in our model and 288 in the AFPSDb model. The reason for this difference is that when the AFPSDb was enlarged to cover more than 200 million proteins, the options to generate the predicted structures were not as fine as those available to a user of the AlphaFold2 software v2.3.1 (in particular, the options associated with templates in the PDB and with the construction of the MSA) that are applied to predict the structure of a single protein. As a further test of agreement between both models, we computed their moments of inertia that resulted in essentially indistinguishable values (amu Å2): 20.308, 33.028, and 35.519 for our model, and 20.202, 33.233, and 35.532 for the AFPSDb model.

In what follows, all results refer to the model of the NcENA1 protein here obtained with Alphafold2. This model, shown in Figure 3, demonstrates structural conservation with P-type ATPases, clearly differentiating the three cytosolic domains typical of this family of proteins corresponding to the A, N, and P domains and the transmembrane domain composed of 10 transmembrane helices (Figure 3).

As mentioned in Materials and Methods, the number of residues with pLDDT values indicating “very high” (AlphaFold term) confidence (466 out of a total of 1077) is particularly high (Appendix A). The MSA showed that many sequences (>17,500) were used for the predictions, most of which have a low identity of approximately 30%, corresponding to sequences of the P2-type ATPase subfamily for which crystallographic data are available (Appendix A). This observation reinforces the reliability of the method. Despite the high number of sequences contained in the MSA, there were sequence segments that had low coverage and that coincided with decreases in per-residue pLDDT values (Appendix A). These segments corresponded with those of the amino acid sequences most specific to P2D-type ENA ATPases that distinguish them from the other ATPases of the P2 subfamily, so the confidence of the prediction was diminished by not having enough reference crystallographic structures. Despite these local minor issues, the overall Alphafold2 metrics indicated that the modeled structure for the NcENA1 protein would be reliable, allowing further analyses of molecular properties such as electrostatic potentials as well as the prediction of possible Na^+^ or K^+^ ion binding sites.

### 2.4. Electrostatic Potentials and Possible Na^+^ and/or K^+^ Binding Sites of NcENA1

Electrostatic potential (EP) maps describe the spatial charge distributions of proteins, providing valuable information for assessing their interactions with other molecules, with ligands or ions, and with the aqueous environment. Electrostatics is the major force intervening in molecular interactions due to its long-range effect since attractions or repulsions arising from atomic charges decay with the inverse of the interaction distance. Furthermore, in the specific case of ion-moving transmembrane proteins, EPs also allow us to evaluate the influence of charged residues involved in the conductance of pathways followed by ions to cross membranes [28]. The PB-EP mapped onto the molecular surface of the NcENA1 protein is shown in Figure 4A.

Note the positive electrostatic potentials of both ATP binding sites and membrane binding sites (Figure 4A). Electrostatic interactions help bind ATP in its binding site and anchor the protein correctly to the membrane by interacting, respectively, with the negative phosphates of ATP and with the negatively charged polar heads of the plasma membrane lipids. On the other hand, Figure 4B shows the PB-EP mapped onto the surface as seen from the cell exterior, in which a large surface with negative PB-EP stands out. This negative potential extends across the entire extracellular surface (Figure 4A, B) and coincides with the ends of the TM helices, thus acquiring great importance in the conductance of the ion channel through an electrostatic mechanism.

However, the ion exit site does not extend across this entire surface; rather, and according to previous studies on other P2 type P-ATPases [29], the ion channel seemed to be formed in the space between helices TM4, TM5, TM6, and TM8. To verify that ENA proteins retain a similar ion channel, we calculated the electric field lines, which indicate the path followed by a free electric charge moving in an electric field, as would be the case for Na^+^ or K^+^ ions. Examining the electric field lines of the extracellular part, shown in Figure 5, it can be observed how there are different zones with a high density of field lines, which indicates a greater magnitude of the electric field in these zones.

One of the areas with high line density is the space formed between helices TM4, TM5, TM6, and TM8, indicating that ENA-type ATPases could conserve the same ion channel as other P-ATPases. However, there are other areas with high line density that, without further analysis, cannot be ruled out initially and might also be involved in the transport of ions across the membrane. To test whether these zones could be part of the ion pathway, two-dimensional slices of the PB-EP map were analyzed across the transmembrane helices along the axis perpendicular to the membrane. If they were part of the ion pathway, an electrostatic potential should be visualized along the entire transmembrane, through which Na^+^ or K^+^ ions could be guided. In the slices shown in Figure 6, it can be observed that at the cytoplasmic border of the membrane there are several points with a negative PB-EP, in particular those between helices TM6, TM7, and TM8, and those between TM4, TM5, and TM6 (Figure 6C), points that could indicate the entry of two Na^+^ and/or K^+^ atoms.

As one dives into the membrane, both the extent of the negative areas and the intensity of the negative PB-EP increase, thus creating a wide zone of negative potential. This would be instrumental for the conductance of positively charged ions (Na^+^ or K^+^) through the ion channel guided by a purely electrostatic mechanism. Therefore, after analyzing the electric field lines on the cell exterior and the electrostatic potential map slices, it appears that, despite having several zones with a high density of electric field lines, the ion channel of the ENA proteins can be formed by the TM4, TM5, TM6, TM7, and TM8 helices.

The different crystallized structures of endoplasmic reticulum calcium pumps (P2A-type ATPases) have allowed the identification of two Ca^2+^ binding sites: site I is composed of TM5, TM6, and TM8 helices residues, and site II is predominantly composed of TM4 residues [29]. The results just presented concerning the finding of an intense negative PB-EP between helices could indicate that the ENA proteins also conserve these binding sites. However, in addition to those four TM4, TM5, TM6, and TM8 helices, it is possible that some residues of the TM7 helix could also be part of the binding sites.

To explore possible Na^+^ or K^+^ binding sites in the predicted structure for NcENA1, two Na^+^ cations were added in the center of the space left between TM4, TM5, TM6, TM7, and TM8 helices, followed by a potential energy minimization of the full protein-ions complex (see Material and Methods). The result of this minimization arranged the two sodium ions 5.6 Å apart from each other and placed them at precise locations that define two binding sites (Figure 7).

Site I would be formed by residues H814 and E818, belonging to the TM5 helix, and T855, S856, and D860 of the TM6 helix, all residues having electronegative oxygens that would provide the negative electrostatic potential. In particular, the two negatively charged acidic residues E818 and D860 would adopt a clamp conformation, maximizing their electrostatic attraction to the cations. Site II would be formed by the main chain oxygen atoms of I341, P342, A343, and S344, residues that form the conserved sequence motif “IPAS” of the TM4 helix, and the side chains of N819 and Q822 of TM5, both residues providing side chain electronegative oxygens.

A closer examination of the amino acid sequences of transported cation binding regions described in other P2-type ATPases for which crystallographic data are available compared with that of NcENA1 revealed clear differences in the residues involved (Appendix A). These differences were interesting and encouraged us to study the residues that might be implicated in the Na^+^ and/or K^+^ binding sites, as well as the specific channel through which they were transported.

### 2.5. Identification of Relevant Functional Residues in NcENA1 ATPase by Site-Directed Mutagenesis

To find out whether the proposed Na+ binding sites could be involved in the binding and transport of the cation across the membrane, some mutants were obtained (Table 1), and the effect of replacement of these amino acid residues on the function of Na+ transport was examined by heterologous expression of NcENA1 mutant proteins in the B3.1 strain of *S. cerevisiae*.

This yeast mutant is very sensitive to Na^+^ and does not grow in the presence of salt in the medium because it lacks its ENA ATPases and the plasma membrane Na^+^-H^+^ antiporter NHA1 (see Materials and Methods). The Na^+^-extrusion activity of the obtained NcENA1 mutants can be inferred from the level of salt-sensitivity complementation of the transformed B3.1 cells expressing the mutant genes. As shown in Figure 8 and Table 1, transformants expressing mutants A343E of TM4, S856A and D860V of TM6, and S963A of TM8 showed reduced salt tolerance compared to the wild-type NcENA1 gene.

However, interestingly, a complete loss of function was observed in the mutants E818V, N819V, or Q822V in TM5, T855V or D860S in TM6, and D892V in TM7, as no growth of the transformants was detected when inoculated in the presence of the lowest salt concentration tested of 20 mM NaCl or 0.2 M NaCl in AP minimal medium or in the complete YPD medium, respectively, as was the case in the negative control expressing the empty vector pDR195 (Figure 8). Therefore, these results suggest the participation of residues E818, N819, Q822, T855, D860, and D892 present in TM4, TM5, TM6, and TM7, respectively, in Na^+^ binding and in the formation of the channel through which this cation moves to the cell exterior, corroborating the prediction made from the above-presented analyses of the NcENA1 model structure.

On the other hand, virtually no differential effect on salt tolerance was observed between cells expressing the S344G mutant and cells harboring the wild-type NcENA1 gene, indicating that not all changes in terms of electrical charge or steric bulk of residues involved in the cation efflux channel abolish the function of the protein.

Although the NcENA1 ATPase from *N. crassa*, unlike other ENA ATPases, has been shown to be very efficient in Na^+^ and not so efficient in K^+^ efflux [17], here we also studied the tolerance to high KCl concentrations of the mutants obtained. Surprisingly, we found that the Q822V, D860V, and S963A mutants showed better growth in the presence of 1.2 M KCl than the wild-type protein (Figure 8 and Table 1), suggesting that these amino acids could participate in the selectivity of the cation to be transported by the protein.

## 3. Discussion

### 3.1. Emergence of a New Clade of P-Type ATPases, Probably Involved in Na^+^ Transport

In this work, we have tried to deepen the phylogeny and 3D structure of ENA-type ATPases, proteins whose function has been studied but not these aspects that have remained largely unknown. An HMM profile has been constructed that have proved to be very useful for the identification of ENA ATPases in the tested databases. This profile fits perfectly for new ENA ATPases identified in databases belonging to eukaryotic organisms of various phylogenetic groups not known so far. Among the phylogenetic groups in which new ENA candidates have been identified are some phylogenetically primitive fungi of the phylum Zoopagomycota, together with others considered *incertae sedis* (an expression indicating the difficulty of exact taxonomic location). In addition, new ENA proteins have been identified in organisms of phylogenetic groups other than fungi, such as the Heterobolosea, Heterokonta, Rhizaria, Bryophytina, Marchantiophyta, and Chlorophyta groups. This work has therefore allowed us to broaden our knowledge of the phylogenetic groups that possess these P2D-type P-ATPases, extending the existence of ENA ATPases to more phylogenetically distant groups of organisms, which apparently do not ascribe to specific metabolic groups (including examples of chemoheterotrophs and photosynthetic organisms) nor to specific ecosystems of life (there are examples of marine organisms, soil dwellers, plant and animal symbionts, etc.).

The phylogenetic analysis carried out here has also made it possible to verify that some proteins initially proposed in the literature as ENA ATPases do not seem to belong to this group of P-ATPases since, although they show a certain phylogenetic proximity, they form a separate clade. In this regard, a recent paper describes the identification of a new family of P-ATPases present in some bacteria, which are not assigned to any of the five P1-P5 families and which have constituted a new clade called P6 [30]. Similarly, our results could suggest that the P2-type ATPase subfamily could be divided into more groups than are currently considered. This new group would be formed mainly by proteins belonging to various groups of protists, including the Apicomplexa, in which *Plasmodium* is located. Having shown that at least some of these ‘non-ENA’ proteins also transport Na^+^, as in the case of PfATP4 from *Plasmodium* [24,31], it could be deduced that the same function of Na^+^ release can be fulfilled by proteins following similar mechanisms of action (both groups of proteins are P-ATPases), but with variations in their physiological conditions of activity, such as pH optimum, physicochemical characteristics, or in the mechanism of binding and extrusion of ions. Therefore, further functional studies would be necessary to expand the characteristics of this new group and find the differences with the other groups of P2-type ATPases. In addition, it would be very interesting to find out what has determined throughout evolution that organisms select the possession of an ENA-type ATPase or other ATPases that show some resemblance to the Na and K ATPases present in animals, which could be responsible for mediating the ionic adjustment of Na^+^ (and/or K^+^) and which apparently do not coexist in any organism so far studied. This interesting study is beyond the scope of this work.

### 3.2. Structure of ENA ATPases and Proposed Na^+^ Binding Sites

In this work, a detailed study of the structure of ENA proteins has been addressed, taking as an example the ATPase of *N. crassa* NcENA1. For this purpose, the most advanced artificial intelligence/deep learning method for protein structure prediction made public in July 2021, AlphaFold2 [26], was applied. ENA protein modeling studies had previously been performed for the yeast *S. cerevisiae* [17] and the moss *Physcomitrella patens* [32] homologous proteins, and, like the structure predicted for NcENA1, they retain the structural relationship with the P-type ATPases, clearly differentiating the three cytosolic domains and the transmembrane domain typical of this protein family (Figure 3). The progress achieved in this work is that through the study of electrostatic potentials, it has been possible to propose the route that Na^+^ follows inside the protein when it is transported to the outside of the cell, and some amino acids have been identified as candidates to participate in this cation transfer as well. Although there is a large surface region with a negative electrostatic potential in the extracellular part of the protein, the negative electrostatic potential in the space inside the TM helices would allow us to propose that the ion channel of the ENA proteins would be formed by the TM4-TM8 transmembrane helices. Analysis of the amino acid composition of these TM helices has allowed us to propose two binding sites for Na^+^ or K^+^ ions formed between TM4-6. Interestingly, the results of targeted mutagenesis performed in the laboratory would support this proposal (Table 1). A mutational study of residues identified as candidates to be important for Na^+^ binding and transport across the membrane to the cell exterior has shown that the removal of negative charges from the side or main chains of these amino acids suppresses to a greater or lesser extent the Na^+^ transport activity, which is deduced by the growth deficit of yeast transformants expressing these mutated proteins in the presence of certain concentrations of NaCl and KCl in the medium (Figure 8). Of all the mutated amino acids, E818, N819, and Q822 in TM5, T855 and D860 in TM6, and D892 in TM7 produced total loss of Na^+^ efflux activity at the lower NaCl concentration assayed (20 mM), suggesting a special role in the ENA ATPase function. Interestingly, residues N819, T855, and D892 are quite conserved among ENA ATPases, so the functional relevance observed here could probably extend to the whole group of ENA ATPases. In fact, the functional relevance of the aspartate in the ENA ATPase of the fungus *Zygosaccharomyces rouxii* equivalent to D892 in NcENA1 has been previously demonstrated [33]. However, the fact that some changes only slightly reduced and did not completely abolish the Na^+^ efflux function does not exclude that these residues also play a role in forming the electrostatic potential that guides the Na^+^ efflux or the entry pathway to the Na^+^ binding sites. In the case of the D860 mutant, it is striking that depending on the substitution made (aspartate to valine or serine), differences in the decrease in transport function were observed, with the inability of the transformant to grow at low NaCl concentrations being very evident in the case of the D860S mutation and more subtle in the case of the D860V (Figure 8). These results would indicate that not only electrical charge, but also steric interactions may be critical in cation binding and transport across the membrane.

Another interesting result reported here concerns the change in selectivity for the K^+^ versus Na^+^ cation of NcENA1 that has been achieved by mutation of certain residues (Figure 8 and Table 1). Specifically, the mutations at D860 and S963, which did not demonstrate a marked phenotype for Na^+^ transport, have nevertheless been shown to be critical for enhancing K^+^ transport. Na^+^ and K^+^ are structurally and chemically very similar elements, although the effective ionic radii vary in isolated cations, with Na^+^ being smaller than K^+^. However, in biological processes, estimates of the radius of hydrated Na^+^ have given values larger than those of K^+^ [34], a result that makes the ionic channels for both cations not equivalent. In the case of the three mutants in which an improvement in K^+^ transport was observed, the amino acid changes were made by introducing one of smaller size (aspartate to valine in D860; serine to alanine in S963; and glutamine to valine in Q822, respectively), which implies leaving more space available for ion transport. But at the same time, they involve removing polarity/charge in the channel, an effect that should not favor the transport of either free or hydrated ions. The size change should favor transport of the larger ion regardless of whether it is free or hydrated, so if the result of the three mutations is that they favor K^+^ transport, this suggests that the ions are not hydrated in the passage through the channel. For the time being, we have no data to support this hypothesis. The cornerstone of the composition of selective filters for Na^+^ and K^+^ in ion channels and transporters remains an unresolved issue. In any case, it will be very interesting to study other ENA proteins, which are more promiscuous in terms of the cation transported or that are specific for K^+^, such as CTA3 from *S. pombe* [16], and the residues they possess in the equivalent proximal positions of the proteins.

In summary, and in the absence of further structural data to confirm these proposals, it could be assumed that ENA proteins would retain the same mechanism of action as other P2-type ATPases. According to this, two atoms of Na^+^ (or probably also K^+^ in the case of ENA ATPases, more specific for that cation) would bind to the two binding sites proposed in this work by electrostatic interactions and would subsequently be extruded from the cell through a negative charge-directed ion channel formed between the TM4-TM8 helices.

### 3.3. Targeted Mutagenesis and Yeast Complementation Studies as a Useful Strategy for Advancing in the Structure-Function of ENA ATPases

In this work, functional studies have been carried out to determine the relevance of certain residues of ENA proteins to Na^+^ transport activity. For this purpose, a simple and versatile method based on the study of the salinity tolerance of yeast transformants expressing mutant proteins has been used. So far, the appropriate way to study the role of a residue in the function of an ATPase has been to test the ATPase enzymatic activity of the mutant protein from purified membranes, in which the release of Pi from ATP as a substrate is determined [35,36] or by performing electrophysiological studies [7]. This involves the use of longer and more complex protocols, which require the use of very specific equipment and skills for their development. Here we have used a screening method in which the ion transport function of the ATPase is quantified instead of the ATP hydrolysis enzymatic activity of the protein. Using the “drop test” analysis, the ability to recover the growth of a highly salt-sensitive yeast mutant by expression of mutant proteins was studied. An improvement in growth in the presence of high NaCl or KCl concentrations implies the recovery of the Na^+^ or K^+^ efflux function through the mutant NcENA1 transporter. This method has been shown to be sensitive enough to be able to finely tune the Na^+^ and K^+^ efflux activity (and thus salt stress relief) that correlates with the growth capacity at high salt concentrations. We propose it as a useful screening method for the activities of this group of proteins as a preliminary approach to further advance this study of the structure-function relationships of ENA ATPases.

## 4. Materials and Methods

### 4.1. Construction of the Profile of the Hidden Markov Model

From a Multiple Sequence Alignment (MSA) of 152 ENA protein sequences extracted from a work described previously [17], created with CLUSTALW software, v2.1, last accessed: 18 October 2023 [37], and subsequently manually revised with the JalView alignment editor [38], a Hidden Markov Model (HMM) profile was created with the ‘hmmbuild’ tool in HMMER3 [21] using the default parameters. After creating the HMM profile of the ENA proteins, it was tested against a group of well-known ENA sequences (not included in the MSA with which the HMM profile was created), as well as against a group of P-type ATPase sequences other than the ENA proteins (called ‘non-ENA’). The results showed that such a profile was very specific since, for the group of ENA sequences, an ‘e-value’ of 0 was obtained for all cases. The e-values for the rest of the ‘non-ENA’ P-ATPases ranged from 1 × 10^−100^ to 1 × 10^−200^ and never reached 0. The very low ‘e-value’ values of the ‘non-ENA’ P-ATPases can be explained by the sequence relationship of all P-type ATPases, which share highly conserved sequence regions that lead to a good score in this test.

### 4.2. Construction of Phylogenetic Trees

The phylogenetic study of ENA proteins was performed with a total of 72 sequences (see Appendix A). Their alignment was performed with the ‘hmmalign’ tool of HMMER3 [21] and then manually edited with JalView [38]. Such MSA was trimmed with TrimAI software, v1.2, (last accessed: 18 October 2023) [39] using the ‘strictplus’ algorithm. The selection of the best evolutionary model for such alignment was performed with ProtTest 3 [40], which indicated that LG + G was the best fitting surrogate model for both alignments. As the BEAST software, v1.10.4, (last accessed: 18 October 2023) [41] does not contain such a model, the second-best model, WAG, was used. Finally, the construction of the phylogenetic tree was performed by Bayesian inference with BEAST software, v1.10.4, with a strict clock. The results were analyzed with Tracer [42], and subsequently, TreeAnnotator [41] was used to obtain the consensus tree. Finally, the consensus tree was edited using the FigTree program (http://tree.bio.ed.ac.uk/software/figtree/, version v1.4.4, last accessed: 18 October 2023).

For the phylogenetic tree of all P-ATPase subfamilies (P1-P5), 48 sequences were chosen (see Appendix A), and the previous process was repeated, changing only the way in which the alignment was constructed, for which CLUSTALW software, v 2.1 was used.

### 4.3. Protein Structure Prediction and Three-Dimensional Modeling of the NcENA1 Protein

Three-dimensional structure prediction of ENA proteins was performed using AlphaFold2 [26]. For this purpose, the ENA1 protein from the fungus *Neurospora crassa* (NcENA1), with NCBI identification code CAB65298.1 [43], was used as an example. Prediction were performed using an unmodified version of ColabFold [44] with the default MSA study, employing a search with MMseqs2 [45] from UniRef [46] and metagenomic sequence databases for this purpose. The AlphaFold2 option to use structures from the PDB (Protein Data Bank, https://www.rcsb.org/, last accessed: 31 October 2023) [47] as templates was selected. The program returned a total of five structure models for the NcENA1 protein, together with their associated metrics. Three relevant metrics provided by AlphaFold2 are: (i) the multiple alignment of homologous sequences, a key issue for the high reliability of the algorithms used by AlphaFold2 since an MSA with a higher number of sequences with good identity with the query sequence yields more reliable structural models; (ii) the set of per-residue values of pLDDT (predicted Local Distance Difference Test score); and (iii) the global pLDDT for each model. This Alphafold2 metric is a measure in the form of a percentage of the confidence of the predictions, with values above 90 indicating a reliability equivalent to that of an experimental structure [26,27].

After the latest release of the AlphaFold Protein Structure Database (AFPSDb), which contains over 200 million predicted structures [48] that provide broad coverage of UniProt, at the time of this writing, we found out that the NcENA1 model is included in the AFPSDb (entry Q9UUX7, code for its sequence in UniProt). However, as discussed in Results, the NcENA1 structure in this database and that obtained by us with the AlphaFold2 software, v2.3.1, last accessed: 18 October 2023, are virtually identical. Therefore, after checking that the set of per-residue pLDDT numerical values agree in both models (a more stringent test of similarity extended over the full structure presented in the Results section) and that the computed moments of inertia (a physicochemical property highly sensitive to geometry details) are nearly indistinguishable in both models, one can assure that all the results discussed in the computational structural analyses presented below do not depend on which of the two AlphaFold models is used.

Since the model structure predicted by AlphaFold2 was to be introduced into a membrane model, it was considered convenient to relax amino acid contacts by optimizing the geometry of the full model prior to such modification. For this, the potential energy was minimized with the ‘Minimize Structure’ tool of Chimera [49]. A total of three gradient-guided minimization cycles were performed: the first with 100 steps of the Steepest Descent (SD) algorithm and 10 steps of the conjugate gradient (CG) algorithm, the second with 200 SD and 20 CG steps, and the third with 500 SD and 50 CG steps. This minimization process was then repeated when the two Na^+^ cations were added to address the binding site in the TM helices. For the computational construction of the lipid membrane, the modeling tool ‘Membrane Builder’ from the CHARMM-GUI service was used [50]. Following the amounts of lipids observed in the literature [51], the model membrane was composed of a 2:1:1:1 ratio of ergosterol, dipalmitoylphosphatidylcholine (DPPC), 1-palmitoyl-2-oleoylphosphatidylserine (POPS), and dipalmitoylphosphatidylserine (DPPS), respectively.

### 4.4. Calculations of Poisson-Boltzmann (PB) Electrostatic Potentials (EPs)

The ‘Adaptive Poisson-Boltzmann Solver’ (APBS) program [52] was used to obtain the PB-EP of the NcENA1 protein. This program solves numerically the Poisson-Boltzmann differential equation using finite element methods through an iterative process in which fine local grids of space points are refined from initial coarse grids. We used the APBS program implemented as the ‘APBS Electrostatics’ plugin PyMOL (https://pymol.org/2/, v2.5.2, last accessed: 18 October 2023). For this purpose, the predicted structure was prepared for APBS computation using the PDB2PQR tool [53], implemented in Chimera. This tool generates PQR-formatted files containing the atomic charges and radii of all the atoms in the structure, which are necessary for the calculation of the PB-EP. APBS calculations were carried out using the nonlinear form of the Poisson-Boltzmann equation, a solvent probe radius of 1.4 Å to simulate water, and a surface sphere density of 10 grid points/Å2. The temperature was set at 298.15 K, the ionic strength at the standard 0.15 M salt concentration of sodium chloride, and the dielectric constants were set at 4 for protein and 78.54 for water. Images of PB-EP mapped onto molecular surfaces were designed and rendered after analyzing the results with PyMOL (https://pymol.org/2/, v2.5.2) and VMD (https://www.ks.uiuc.edu/Research/vmd/, v1.9.3) softwares, (last accessed: 18 October 2023).

### 4.5. Site-Directed Mutagenesis of NcENA1 ATPase

In this work, an inverse PCR mutagenesis procedure was carried out using the NcENA1 cDNA cloned in the TOPO TA cloning vector (Invitrogen) as a template and two primers oriented in the reverse direction that contain the changes in bases of interest and 5’phosphorylated ends to allow the two ends to be ligated together following amplification and reconstituting the entire plasmid with the mutations included [54]. The sequences of the primers used for the site-directed mutagenesis of all constructs are listed in Appendix A. After PCR, *Dpn*I digestion for 1 h at 37 °C was carried out to cleave the template, which contained methylated sites. The resulting plasmid was then transformed into *Escherichia coli* competent DH5α cells for propagation. All PCR products were sequenced to check for introduced mutations. Finally, to study the function of the protein encoded by the mutated cDNA, it was cloned into the *Not*I site of the polylinker of the yeast expression vector pDR195 [55].

### 4.6. Yeast Transformation for Phenotypic Characterization of the NcENA1 Mutants

The functional characterization of the NcENA1 mutants was performed in the *S. cerevisiae* mutant strain called B3.1 (Mat *a ade2 ura3 trp1 ena1-4::HIS3 nha1::LEU2*), in which the Na^+^ efflux systems ENA1-4 and NHA1 are absent [56]. This yeast mutant is very sensitive to salinity because it is unable to extrude excess Na^+^ that accumulates in this condition and does not grow in the presence of high salt concentrations. Thus, the characterization of the transport activity of the NcENA1 mutants was tested by studying the total or partial ability of the mutant proteins to reverse the growth of the yeast transformants in the presence of high concentrations of NaCl and/or KCl. For yeast transformations, the lithium acetate/PEG 4000 cell permeabilization method was used [57]. All the constructs of NcENA1 mutants were cloned and expressed in the pDR195 shuttle vector under the PMA1 promoter [55]. Yeast strains were normally grown either in complex yeast-peptone-dextrose (YPD) medium (1% yeast extract, 2% peptone, and 2% glucose) or in minimal SD medium [58]. Growth at variable K^+^ and Na^+^ concentrations was conducted using arginine phosphate (AP) medium [59] or YPD medium supplemented with the indicated KCl and NaCl concentrations.

### 4.7. Salt Tolerance Assay of Yeast Transformants with Mutated NcENA1 Constructs

For the functional characterization of the ENA mutants, drop test analysis was carried out with the yeast transformants expressing the different NcENA1 mutants. For that, tenfold serial dilutions of suspensions of the yeast transformants expressing the NcENA1 mutant were prepared in 96-well plates. Using a flame-sterilized 48-pinner replica plater designed for 96-well plates, drops of all the yeast transformant dilutions were transferred at once to different solid YPD or AP medium plates containing different NaCl and KCl concentrations. The growth of the different transformants was observed after four-six days of incubation at 28 °C. A yeast transformant expressing the pDR195 empty vector was used as a negative control. The drop-test assays of the NcENA1 mutants were performed in triplicate using two to four independent transformants expressing each NcENA1 mutation. The results obtained were identical between the replicates, and Figure 8 shows an example for each mutant of the growth under the different saline conditions.

## Figures and Tables

**Figure 2 ijms-25-00514-f002:**
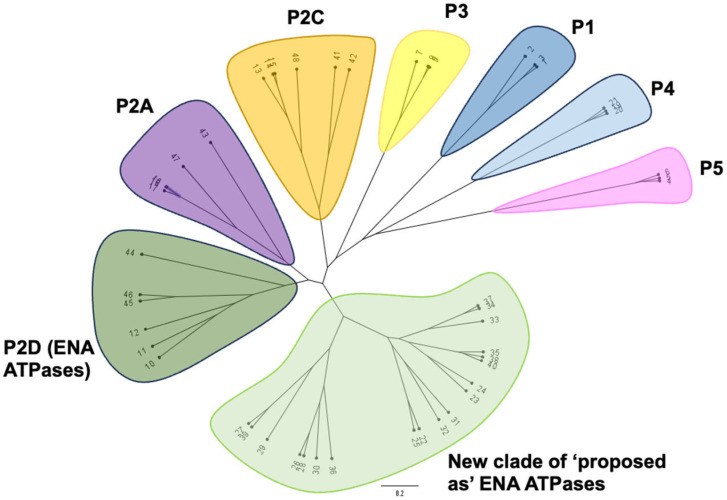
A phylogenetic tree showing the main clades of P-type ATPases. Its construction was based on the alignment of 48 proteins representative of the main clades compiled in Appendix A (Appendix A). The main classes of ATPases are distinguished by different colors: ENA ATPases (P2D, in dark green), SERCA ATPases (P2A, purple), Na/K ATPases (P2C, orange), H^+^ ATPases (P3, yellow), heavy metal ATPases (P1, dark blue), ER ATPases (P5, pink), and phospholipid ATPases (P4, light blue) groups are highlighted. The new clade of ‘proposed as’ ENA ATPases is marked in light green.

**Figure 3 ijms-25-00514-f003:**
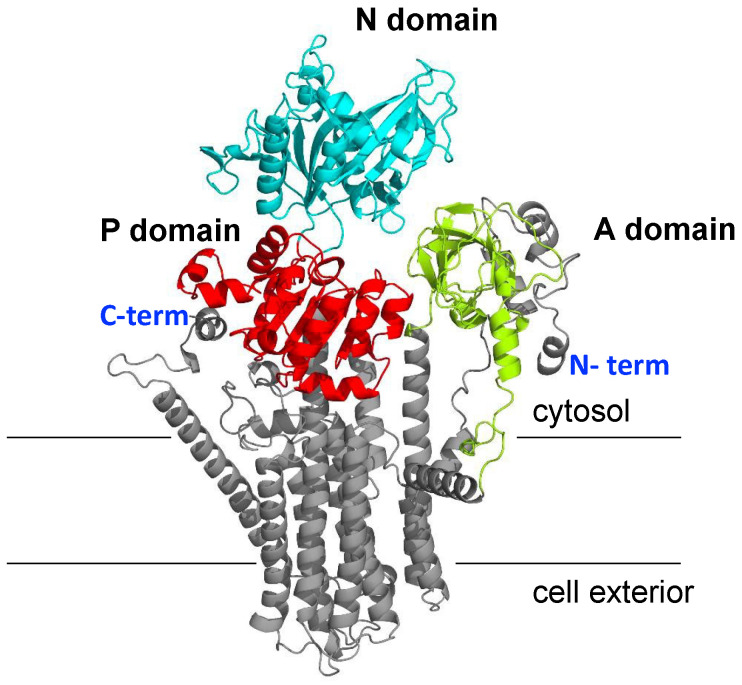
Three-dimensional structure model for the NcENA1 protein obtained with the AlphaFold2 software v2.3.1. The model is colored according to the cytosolic domains: A domain (lime green), P domain (red), and N domain (cyan). The rest of the protein is colored gray, corresponding to the TM helices and N- and C-termini of the protein. The lines indicate the approximate limits of the lipid membrane.

**Figure 4 ijms-25-00514-f004:**
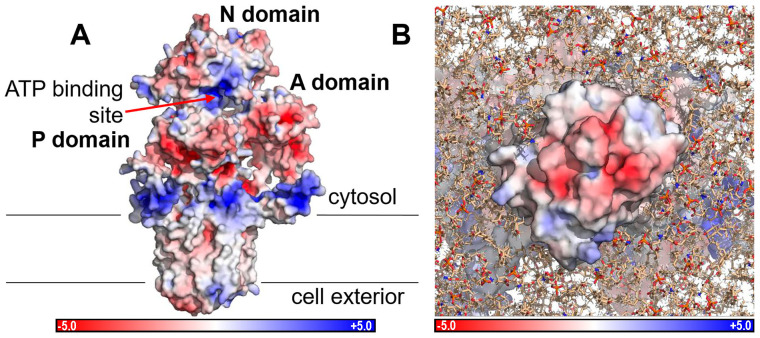
Poisson Boltzmann electrostatic potential (PB-EP) mapped onto the molecular surface of the complete NcENA1 protein ((**A**), side view) and of the extracellular part ((**B**), top view). Red colors indicate a negative PB-EP, while blue colors indicate a positive PB-EP according to the color bar below, whose units are kT/e (1 kT/(eÅ) = 2.59 × 108 V/m). The NcENA1 molecule is shown in the same orientation as in Figure 3. Lines in (**A**) indicate the approximate limits of the lipid membrane. The lipid membrane in (**B**) is constructed with the tools available in the CHARMM-GUI web (see Material and Methods in Section 4).

**Figure 5 ijms-25-00514-f005:**
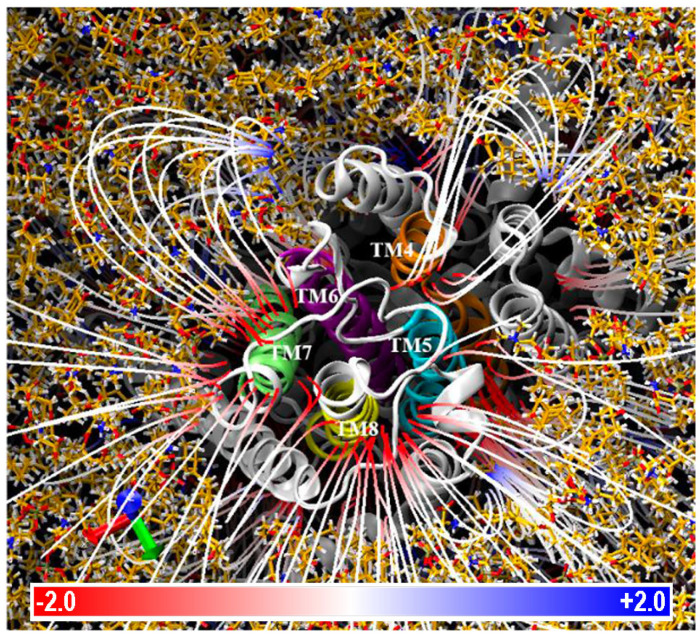
Electric field lines of the extracellular interface of the NcENA1 protein. Lines are shown on a scale of −2.0 to 2.0 kT/Å according to the color bar. TM4 (orange), TM5 (blue), TM6 (violet), TM7 (lime green), and TM8 (yellow) helices are highlighted. Note the high concentration of field lines in the space between the highlighted TM helices. Lipids (sticks with yellow carbons) of the membrane set using CHARMM-GUI as explained in Material and Methods are shown surrounding NcENA1 (gray cartoon).

**Figure 6 ijms-25-00514-f006:**
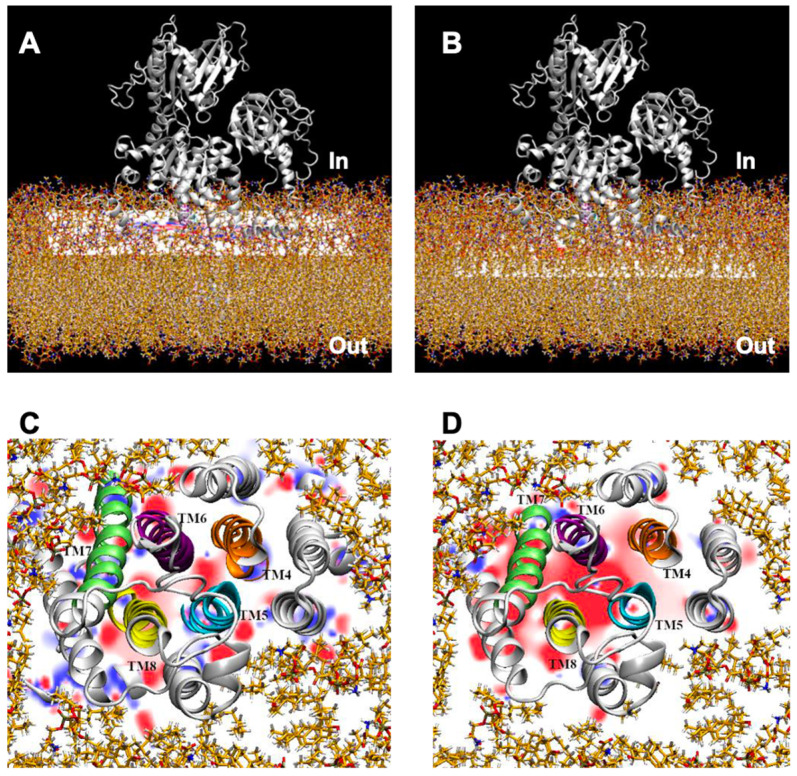
Two-dimensional slices of the PB-EP cutting transmembrane helices. TM4 (orange), TM5 (blue), TM6 (violet), TM7 (green), and TM8 (yellow) along the axis perpendicular to the membrane at the cytoplasmic border (**A**,**C**) and at the middle of the membrane (**B**,**D**). Red and blue colors in the slices indicate negative and positive PB-EPs, respectively, on a scale ranging from −20 to +20 kT/e.

**Figure 7 ijms-25-00514-f007:**
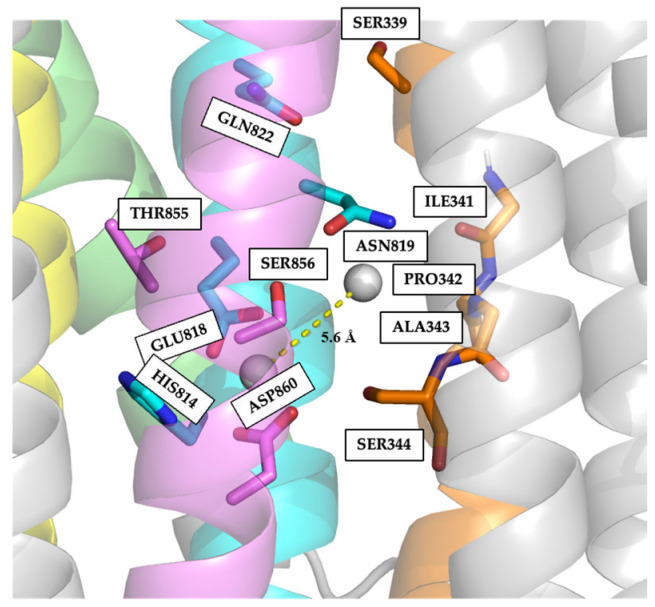
Possible Na^+^ binding sites of NcENA1. The TM4 (orange), TM5 (blue), TM6 (violet), TM7 (green), and TM8 (yellow) helices are highlighted, as are the side chains of the following residues: H814, E818, N819, and Q822 in TM5; T855, S856, and D860 in TM6; and S339, I341, P342, A343, and S344 in TM4. Na^+^ ions are represented by the silver balls, separated by a distance of 5.6 Å, shown as a yellow dashed line.

**Figure 8 ijms-25-00514-f008:**
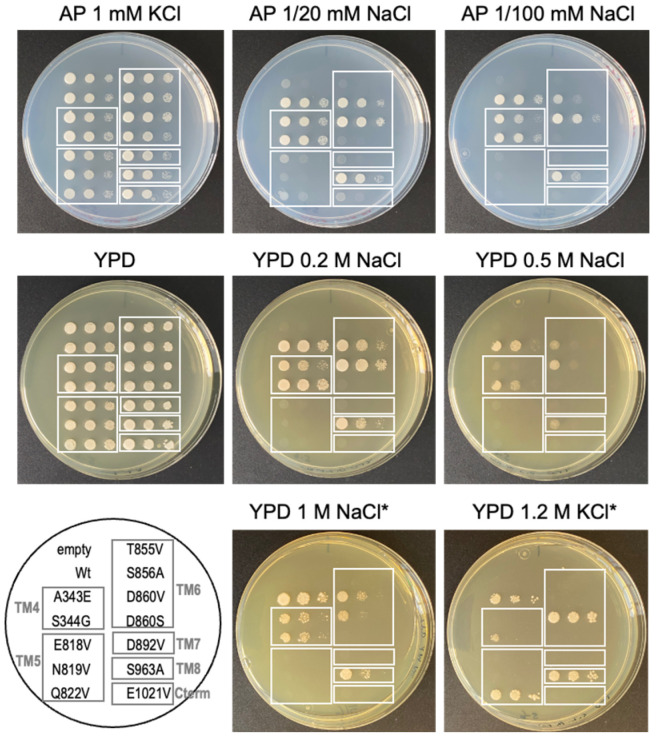
Drop test assay of B3.1 transformants expressing different NcENA1 mutants growing at different NaCl and KCl concentrations. NcENA1 mutants at specific residues proposed as putative Na^+^ binding sites relevant for the protein activity were obtained as described in Materials and Methods. The corresponding cDNAs were cloned into the pDR195 vector and expressed in yeast mutant B3.1. The phenotype was assayed by testing the growth capacity of the transformants in the presence of different NaCl and KCl concentrations. The photos shown were taken after two days of incubation at 28 °C, except for those plates of YPD 1 M NaCl and 1.2 M KCl (labeled with *) that were incubated for four days. The drop test was performed with three to six independent transformants for each mutation, and an example of the results obtained is shown in the Figure. Within the boxes are grouped the mutated residues that are located in each TM (TM4, TM5, TM6, TM7, and TM8). “C-term” corresponds to a transformant expressing the protein whose mutated residue is outside the TMs and is located at the C-terminus of the protein.

**Table 1 ijms-25-00514-t001:** Changes made in some NcENA1 residues by targeted mutagenesis and their effect on Na^+^ and K^+^ growth when expressed in B3.1 yeast mutants. Wild-type sequences (NcENA1(wt)) are shown in the blue background rows, indicating the transmembrane fragment where they are located. The modified sequences are shown in the white background rows, with the mutated residue in red. Growth in the presence of NaCl and KCl as well as the deduced ion efflux phenotype are described in the last three columns, respectively. Symbols ++, +, +, −, −− are used to indicate the level of growth observed in the culture media ranging from increased growth (++) to total absence of growth (−−). ↑ or ↓ indicate increased or decreased growth relative to the wild type respectively.

Strain Name	Protein Sequence	TM Location	Growth in Na^+^	Growth in K^+^	Comments on Phenotype
**NcENA1(wt)**	**I P A S L**	TM4	**++**	**+**	**Na^+^-efflux/slight K^+^ efflux activity**
**A343E**	**I P E S L**	“	+	−−	↓ Na^+^-efflux/Loss of K^+^ efflux
**S344G**	**I P A G L**	“	++	−	↓ K^+^ efflux
**NcENA1(wt)**	**E N I A Q**	TM5	**++**	**+**	**Na^+^-efflux/slight K^+^ efflux activity**
**E818V**	**V N I A Q**	“	−−	−−	Loss of Na^+^ and K^+^ efflux function
**N819V**	**E V I A Q**	“	−−	−−	Loss of Na^+^ and K^+^ efflux function
**Q822V**	**E N I A V**	“	−−	++	Loss of Na^+^ efflux/ ↑ K^+^ efflux
**NcENA1(wt)**	**T S G L P D**	TM6	**++**	**+**	**Na^+^-efflux/slight K^+^ efflux activity**
**T855V**	**V S G L P D**	“	−−	−−	Loss of Na^+^ and K^+^ efflux function
**S856A**	**T A G L P D**	“	+	−−	↓ Na^+^-efflux/Loss of K^+^ efflux
**D860V**	**T S G L P V**	“	+	++	↓ Na^+^-efflux/ ↑ K^+^ efflux
**D860S**	**T S G L P S**	“	−−	−−	Loss of Na^+^ and K^+^ efflux function
**NcENA1(wt)**	**F I I D M I F Y**	TM7	**++**	**+**	**Na^+^-efflux/slight K^+^ efflux activity**
**D892V**	**F I I V M I F Y**	“	−−	−−	Loss of Na^+^ and K^+^ efflux function
**NcENA1(wt)**	**F L A W E L V D M R R S**	TM8	**++**	**+**	**Na^+^-efflux/slight K^+^ efflux activity**
**S963A**	**F L A W E L V D M R R A**	“	+	++	↓ Na^+^-efflux/ ↑ K^+^ efflux
**NcENA1(wt)**	**W E W G I V**	C-term	**++**	**+**	**Na^+^-efflux/slight K^+^ efflux activity**
**E1021V**	**W V W G I V**	“	−−	−−	Loss of Na^+^ and K^+^ efflux function

## Data Availability

Data is contained within the article and Appendix A.

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
