# Peer review of "Phylogenetic and Structure-Function Analyses of ENA ATPases: A Case Study of the ENA1 Protein from the Fungus Neurospora crassa"

_ijms, 2023, doi:10.3390/ijms25010514_

Round 1

Reviewer 1 Report

Comments and Suggestions for Authors

Using extensive phylogenetic analysis, the authors updated the list of organisms containing ENA-type P-ATPases. This allows for expansion of research on this transportes and thereby, enlarges a group of organisms that could potentially be targeted by pharmacological drags inhibiting ENA P-ATPases.

Instead of purifying and crystalizing a candidate from ENA P-ATPases, the authors used an efficient approach by applying the artificial intelligence deep learning method for protein structure prediction, AlphaFold2. This could be done mainly because P-ATPases are structurally much conserved and several crystal structures are currently available. I appreciate that the authors took an effort to test experimentally the predicted Na+/K+ channel using mutagenesis. The screening method, which probes directly the ion transport function, is simple, fast, and elegantly innovative.

Overall, the study by Aguilella et al. advances the knowledge on ENA P-ATPases, and will find interest among readers of IJMS.

I have a few comments that should to be addressed.

Lines 86-94. ENA-type of proteins are P2D type of ATPase. What is a mechanistic difference between P2D (ENA) and P2C (Na+/K+)? Both clades have Na+ transport capacity. Please, provide some explanatory details to a general reader unfamiliar with ATPases.

Figure 3. This figure does not show where the N and C terminus are. Please place “N” and “C” mark.

Line 404: Figure S1A should be in parenthesis.

Figure 4. Please add to the figure legend that A is a side view of the full NcENA1 and B is a top view of the extracellular portion of the protein.

Figure 3. and Figure4. Is the NcENA1 molecule in the same orientation in Fig. 3 and Fig. 4? If so, please add this information somewhere in the text. This would help to determine the position of the A, N, and P domains in Figure 4 more clearly.

Figure 5. The light green line for the high-density zone is not visible in the figure.

Figure 8., Table 1. Data show one example of the results. A downside of this presentation of data is that a reader cannot see how big a deviation from this result with other independent experiments was. Is there any way to show deviation in independent experiments?

Line 574-576. Can the authors provide their mechanistic hypothesis of how increase in hydrophobicity (Q822V, D860V, and S963A) can lead to K+ selectivity?

Figure 3.-6.The crystal structure of the Sus crofa Na+, K+-ATPase (PDB ID: 4HQJ) shows beta-subunit at the extracellular side. This subunit is not mentioned in Fig. 3-6. Does NcENA1 lack this subunit?

Line 615: I suggest omitting “About” in the subtitle.

Line 629: There seem to be excess space before “Analysis”.

Line 633: Please remove comma after “exterior”.

Line 657: the word “being” and “Na” should be switched in order.

Line 659-662: I do not understand the reasoning behind the hypothesis. Introducing a smaller amino acid residue to a channel should make more a space in the channel for an ion with a larger radius. Therefore, introducing valine or alanine by mutagenesis should facilitate/increase the transport of the larger radius of Na+.

Line 660: Using “smaller” rather than “lower” size may be more appropriate word.

Line 665: There should be comma after “proteins”. It may be better to replace “that” with “which”.

Discussion. Is there any reason for why the font size is smaller in the Discussion section than the font in the Results section?

Comments on the Quality of English Language

Mino changes needed

Author Response

Dear reviewer,

Thank you very much for all your corrections and suggestions that will make a clearly improved version of this manuscript. Please find the detailed responses below and the corresponding corrections highlighted in red in the re-submitted file.

Lines 86-94. ENA-type of proteins are P2D type of ATPase. What is a mechanistic difference between P2D (ENA) and P2C (Na+/K+)? Both clades have Na+ transport capacity. Please, provide some explanatory details to a general reader unfamiliar with ATPases. Thank you very much for your suggestion. A brief explanation of the mechanistic differences between P2D and P2C type ATPases has now been included in lines 86-87 of the revised version of ijms-2771291 (please see attachment file).

Figure 3. This figure does not show where the N and C terminus are. Please place “N” and “C” mark. Thank you for pointing this out. The N-terminal and C-terminal ends are now marked in Fig. 3.

Line 404: Figure S1A should be in parenthesis. Done, thank you!!

Figure 4. Please add to the figure legend that A is a side view of the full NcENA1 and B is a top view of the extracellular portion of the protein. Agree. Now this information has been added in the figure legend.

Figure 3. and Figure 4. Is the NcENA1 molecule in the same orientation in Fig. 3 and Fig. 4? If so, please add this information somewhere in the text. This would help to determine the position of the A, N, and P domains in Figure 4 more clearly. The reviewer is right, the NcENA1 molecule in Figure 4 is shown in the same orientation as in Figure 3 and now it is stated in the legend to Figure 4. Thank you.

Figure 5. The light green line for the high-density zone is not visible in the figure. In fact, the “light green line” mentioned in the legend to Figure 5 is not seen at all. When preparing this figure, we initially rendered field lines with a variety of density ranges, but the results at high densities of lines were images too overloaded in which the helices were nearly completely hidden. We finally decided to present the clearer image seen in Figure 5 but unfortunately, we did not remove the mention to the limit line in light green color: we apologize for this mistake. In the revised version we have modified the legend to this Figure to fix this mistake.

Figure 8., Table 1. Data show one example of the results. A downside of this presentation of data is that a reader cannot see how big a deviation from this result with other independent experiments was. Is there any way to show deviation in independent experiments? The intention of the authors has been to show in Figure 8 the ability of the drop test to reveal the differences in the activity of the NcENA1 mutants under different growth conditions and, therefore, direct results and not statistical data are shown. These assays were performed in triplicate using two to four independent transformants expressing each NcENA1 mutation. The results obtained were identical between the replicates and Figure 8 shows an example of the growth under the different saline conditions for each mutant. This information has now been included in the new version.

Line 574-576. Can the authors provide their mechanistic hypothesis of how increase in hydrophobicity (Q822V, D860V, and S963A) can lead to K+ selectivity? This is an interesting point highlighted by the reviewer. However, at the moment we do not have a clear hypothesis to explain whether it is the change in amino acid hydrophobicity, or size or both that can cause the ion channel selectivity to change, and it is certainly an aspect that we intend to study in the near future.

Figure 3.-6. The crystal structure of the Sus crofa Na+, K+-ATPase (PDB ID: 4HQJ) shows beta-subunit at the extracellular side. This subunit is not mentioned in Fig. 3-6. Does NcENA1 lack this subunit? Yes, it does. The information available so far about ENA-type ATPases is that they are proteins that do not require the expression of any subunit to be active.

Line 615: I suggest omitting “About” in the subtitle. Agree. Now it has been changed.

Line 629: There seem to be excess space before “Analysis”. Thank you again. It has been corrected.

Line 633: Please remove comma after “exterior”. You are right. Comma now has been deleted.

Line 657: the word “being” and “Na” should be switched in order. Sorry for the error and thank you very much for bringing it to our attention. It has been corrected.

Line 659-662: I do not understand the reasoning behind the hypothesis. Introducing a smaller amino acid residue to a channel should make more a space in the channel for an ion with a larger radius. Therefore, introducing valine or alanine by mutagenesis should facilitate/increase the transport of the larger radius of Na+. The reviewer is right in that this issue is not clear enough. As we described, hydrated Na+ has a greater radius than K+ whereas free K+ is larger than Na+. Since the three mutations involve changing amino acids with relatively large charged or polar side chains for smaller nonpolar amino acids, there are two effects associated to the mutations. On the one side, removing charged/polar groups should not favor ion transport neither in the free nor in the hydrated state. On the other side, smaller amino acids should favor transport of the larger ion irrespective of whether it is free or hydrated. Although the evidence available on the state of ions being transported is not conclusive enough, the information provided by the three mutants is concerned might support the hypothesis that transported ions are not hydrated. We have modified the text in lines 683-688 (former 659-662 in the initial version of our manuscript) to clarify this issue.

Line 660: Using “smaller” rather than “lower” size may be more appropriate word. Thank you very much. Now it has been corrected.

Line 665: There should be comma after “proteins”. It may be better to replace “that” with “which”. Thank you very much. Now it has been corrected.

Discussion. Is there any reason for why the font size is smaller in the Discussion section than the font in the Results section? Not at all!!, my apologies. It was a mistake when pasting the text into the journal format. It has been corrected in the new version.

Reviewer 2 Report

Comments and Suggestions for Authors

The manuscript “Phylogenetic and structure-function analyses of ENA ATPases. A case study of ENA1 protein from fungus Neurospora crassa” by Aguilella et al. aimed to study ENA transporters, one of P-type ATPases that are characterized by actively moving Na+ or K+ out of the cell in other higher organisms such as plants or animals. It is an original topic in the field. The authors expanded the study about one type of P-ATPases, ENA protein. Current results provide insights into the existence of these transporters in the different organisms, and may become a potential target in the control of pathogenic protozoa and fungi.

The authors utilized a hidden Markon model and Bayesian inference to identify homologous sequences from 152 ENA proteins, and found a new type of P-type ATPase (non-ENA) from the fungal group. In addition, they predicted the 3D structure of this protein through AlphaFold2, to further analyze its electrostatic potential. Lastly, they verified the potential by site-directed mutagenesis and the expression in the yeast. This study is very interesting, which expands the knowledge about the existence of ENA in organisms belonging to higher phylogenetic groups. 

Overall, the method used in the study is thorough. Conclusions are appropriate and supported by the data, the references are cited appropriately. I recommend this study for publication after minor revision.

Here are some minor concerns shown below:

1. Line 136, incomplete text “.... from [17]”, it seems that there are missing texts.

2. Line 285-286: it might be useful to label the new ENA proteins identified in the eukaryotes by using the asterisk marker in the figure 1. 

3. Line 287-293, figure 1, the authors listed three P2-type P-ATPases as an outgroup, as they mentioned in the figure legend. It is fine. But “Outgroup” is not the phylum name. The reviewer suggests deleting the text “outgroup” in the figure, since they were mentioned in the figure legend. 

4. Line 411: which P2 subfamily does NcENA1 protein belong to? 

5. Line 430: “... face of the complete NcENA1 protein”

6. In figure 4, it might be helpful to label “N domain”, “P domain”, and “A domain” in the figure 4A if it is possible. 

7. In figure 5, please add a color scale of -2.0 (red) to 2.0 (blue) in the figure. 

8. In figure 8, the authors visualized the growth capacity of B3.1 transformants expressing different NENA1 mutants by using a drop test assay. Are there any ways to show the deviations between them? 

9.  Please proofread the whole manuscript and formatting the font size, for example: 

the subtitles are suggested not to be italic font.

Change the font size of discussion section.

Author Response

Dear reviewer,

Thank you very much for all your corrections and suggestions that will make a clearly improved version of this manuscript. Please find the detailed responses below and the corresponding corrections highlighted in red in the re-submitted file.

  1. Line 136, incomplete text “.... from [17]”, it seems that there are missing texts. Thank you, the sentence has been completed in the new version.
  2. Line 285-286: it might be useful to label the new ENA proteins identified in the eukaryotes by using the asterisk marker in the figure 1. 3. Line 287-293, figure 1, the authors listed three P2-type P-ATPases as an outgroup, as they mentioned in the figure legend. It is fine. But “Outgroup” is not the phylum name. The reviewer suggests deleting the text “outgroup” in the figure, since they were mentioned in the figure legend. Figure 1 has been modified according to the reviewer's suggestion, including asterisks in the phylogenetic groups in which new ENA ATPases have been identified and eliminating the term "Outgroup". Many thanks for the suggestions.
  3. Line 411: which P2 subfamily does NcENA1 protein belong to? NcENA1 belongs to type P2D. For clarity, it has been included in the text of the new version.
  4. Line 430: “... face of the complete NcENA1 protein”. Thank you for pointing this out. The legend in Figure 4 has been modified to indicate the orientation of the protein shown in the image. This coincides with the orientation in Figure 3 in which the major domains of the P-ATPases are indicated.
  5. In figure 4, it might be helpful to label “N domain”, “P domain”, and “A domain” in the figure 4A if it is possible. Agreed. The domains of the P-type ATPases have been labeled in the figure.
  6. In figure 5, please add a color scale of -2.0 (red) to 2.0 (blue) in the figure. Done. Thanks for the suggestion.
  7. In figure 8, the authors visualized the growth capacity of B3.1 transformants expressing different NENA1 mutants by using a drop test assay. Are there any ways to show the deviations between them? The intention of the authors has been to show in Figure 8 the ability of the drop test to reveal the differences in the activity of the NcENA1 mutants under different growth conditions and, therefore, direct results and not statistical data are shown. These assays were performed in triplicate using two to four independent transformants expressing each NcENA1 mutation. The results obtained were identical between the replicates and Figure 8 shows an example of the growth under the different saline conditions for each mutant. This information has now been included in the new version.
  8. Please proofread the whole manuscript and formatting the font size, for example: 

the subtitles are suggested not to be italic font. My apologies. It has been corrected in the new version.

Change the font size of discussion section. My apologies. It was a mistake when pasting the text into the journal format. It has been corrected in the new version.

Reviewer 3 Report

Comments and Suggestions for Authors

The results obtained in this paper “Phylogenetic and structure-function analyses of ENA ATPases. A case study of ENA1 protein from fungus Neurospora crassa” are interesting and deserve publication. ENA proteins are not found in higher organisms such as plants or animals. The authors constructed the hidden Markov model profile to identify homologous sequences to ENA proteins in protein databases. This analysis helped the authors to identify the existence of ENA-type ATPases in the most primitive groups of fungi, as well as in other eukaryotic organisms not described so far.

The authors discuss the emergence of a new clade of P-type ATPases likely involved in Na+ transport, the structure of ENA ATPases, and putative Na+ binding sites.

The authors proposed the protein regions and the amino acids involved in the transport of Na+ or K+ ions across the membrane for the first time. Targeted mutagenesis of some of these residues has confirmed their relevant participation in the transport function of ENA proteins.

There may be a longer abstract due to the introductory part, which is more suitable for introduction.

Author Response

Dear reviewer,

thank you very much for taking the time to review this manuscript and make the final work look better. In accordance with your suggestion, we have shortened the abstract, which was initially too long, by moving some of the information to the Introduction, which in the new version appears in red (please see attachment).
